# Spatiotemporal Evolution Characteristics in Ecosystem Service Values Based on Land Use/Cover Change in the Tarim River Basin, China

**Yang Wang [1],* , Shuai Zhang [1], Hui Zhen [1], Xueer Chang [1], Remina Shataer [1] and Zhi Li [2]**

1   College of Grassland and Environment Sciences, Xinjiang Agricultural University, Urumqi 830052, China; GMFzhangshuai@163.com (S.Z.); zhenhuihuizi@163.com (H.Z.); changxueer0317@163.com (X.C.); 13079904349@163.com (R.S.)
2   State Key Laboratory of Desert and Oasis Ecology, Xinjiang Institute of Ecology and Geography, Chinese Academy of Sciences, Urumqi 830011, China; liz@ms.xjb.ac.cn
*   Correspondence: ktwangyang@163.com; Tel.: +86-139-9925-2425

**Abstract:** This paper explores the watershed land use and ecosystem services value (ESV) space-time evolution characteristics in the Tarim River Basin in China's arid northwest. The study applies spatial correlation analysis using Landsat TM remote sensing images for 1990, 2000, 2010, and 2018. The land use data are extracted and the ESV coefficients are adjusted accordingly. The results show as follows: (1) From 1990 to 2018, land use in the Tarim River Basin changed significantly. Construction land, cultivated land, and water exhibited an increasing trend, while grassland, forest land, and water indicated a decreasing trend. Construction land increased the most, while water decreased the most. (2) Overall, ESV in the Tarim Basin charted a downward trend, from 872.884 billion RMB in 1990 to 767.165 billion RMB in 2018. From 2015 to 2018, the Basin's ESV suffered the largest declines, with grassland ESV accounting for over 39% of the loss and adjustment services accounting for over 62%. (3) During the study period, the spatial distribution of ESV in the study area showed spatial distribution characterized that was either high in all directions or low in the middle, with significant positive spatial autocorrelation. The spatial distribution of ESV dynamic changes showed that ESV value-added regions were distributed in the southeast portion of the study area, while the ESV loss regions were distributed in the western and northern portions of the study area.

**Keywords:** land use; ecosystem services value; spatial autocorrelation; Tarim River Basin

## 1. Introduction

Ecosystem services are products and services that are directly or indirectly related to the well-being of people and are acquired through ecosystem structure, process, and function [1]. These types of services form the basis for stable and sustainable development of an ecological environment [2]. Costanza et al. [3] developed the Global Ecosystem Service Value Assessment System in 1997, making ecosystem service a hot research topic in ecology. At present, scholars in China and elsewhere are carrying out extensive research to determine a unified method of ecosystem service value (ESV) assessment. Dawson and colleagues suggested including social complexity as part of ESV assessment [4], while Hassan et al. proposed that there is spatial heterogeneity in ecosystems in different research areas, so the use of the same set of ecosystem service value coefficients in the assessment of different research areas would result in controversy [5].

The Chinese scholars Xie et al. [6] improved the ESV coefficient based on the Costanza [3] evaluation model and the actual situation of China's ecosystem. According to these criteria, they established a set of ESV coefficient that can be widely used to evaluate ecosystem services in Chinese provinces and

cities [7], islands [8], and river basins [9]. At present, ecosystem service value evaluation can be roughly divided into two categories, namely methods based on unit service function prices (functional value method) [10] and methods based on unit area value equivalent factors (equivalent factor method) [11]. The functional value method has more input parameters, which makes the calculation process more cumbersome. The most important thing is that it cannot well unify the evaluation methods and parameter standards of each service value [12]. Compared with the service value method, the equivalent factor method is more intuitive, easy to use, and requires less data. It is very suitable for evaluating the value of ecosystem services on a regional and global scale [13]. The premise of using the equivalent factor method to evaluate the value of ecosystem services is to construct an equivalent factor table. In order to construct a more objective and accurate equivalent factor table, Xie [14] proposed that on the basis of the classification of ecosystem service functions by Costanza et al. [15], a method of valuing ecosystem services based on expert knowledge was constructed. It is widely used to evaluate the value of ecosystem service functions at sample points, regions, and nationwide. Compared to other evaluation methods (e.g., the unit service function price method, the market value method, and the conditional value method), the equivalent factor method has the advantages of simplicity, low data demand, high comparability of results, and comprehensive evaluation, making it widely used in the study of ESV assessment [16–19].

Land use and land cover change (LUCC) is the most fundamental manifestation of the coupling relationship between human activities and terrestrial ecosystems. It is also an important driving factor in the transformation of ecosystem service functions [20]. In the process of land use change, there are changes in the ecosystem service function, structure and process, causing the ESV also to change [21]. However, with the acceleration of social and economic development in recent years [22], unreasonable changes in land cover types have caused serious damage to the ecosystem. Consequently, the value of ecosystem services has also changed, resulting in the degradation of ecosystem service functions [23]. It is therefore critically important to explore the response mechanism of ESV in relation to the evolution of land use patterns for sustainable development and ecosystem restoration.

The Tarim River Basin is located in an arid, ecologically fragile zone in northwest China [24]. It belongs to the core area of the Silk Road Economic belt [25] and is the largest inland river basin in the country. It is also an important cotton production and petrochemical base [26]. In recent years, the Tarim River Basin has been affected by natural and human activities, and the land use patterns have changed dramatically [27]. Numerous ecological problems [28], such as the degradation of natural grassland [29], glacier shrinkage [30], river interruption [31], misallocation of water resources, and accelerated desertification, have become increasingly prominent, making the basin ecological environment more fragile [32]. As a result, many scholars are currently focusing more attention on the sustainable development and restoration of the Tarim River Basin ecosystem [33].

In view of this, the present paper proposes the hypothesis that the ecosystem service value of the Tarim River Basin from 1990 to 2018 shows a loss trend. Further, the research selects land use data from 1990, 2000, 2010, and 2018, based on the ecosystem service value estimation model and exploratory spatial analysis methods, in order to quantitatively explore the impact of land use changes in the Tarim River Basin with regard to temporal and spatial dynamic trends of ecosystem service values. The aim of this research is to provide a scientific basis for the sustainable use of land resources in arid areas and regional ecological environment management decision-making.

## 2. Data and Materials

### 2.1. Description of Study Area

The Tarim River Basin is located in the southern part of the Xinjiang Uygur Autonomous Region (73°10′–94°05′ E and 34°55′–43°08′ N), at the southern foot of the Tianshan Mountains. It is the largest inland river basin in China and encompasses more than 90 counties and cities, along with the

Taklimakin Desert. The Basin has a total area of $1.02 \times 10^6$ km$^2$, accounting for 61.4% of the total area of Xinjiang (Figure 1).

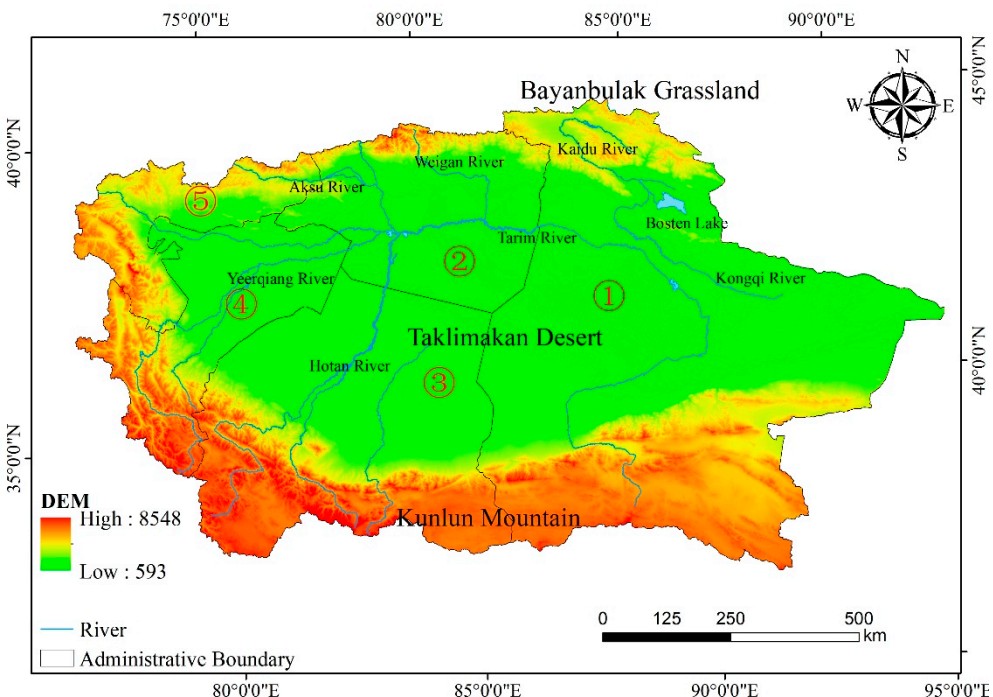

**Figure 1.** Sketch map of study area: 1. Bayingolin Mongolian Autonomous Prefecture; 2. Aksu Prefecture; 3. Hotan Prefecture; 4. Kashi Prefecture; 5. Kizilsu Kirghiz Autonomous Prefecture.

Due to its deep inland location and unique topography, the Tarim River Basin has a dry climate that is characterized as temperate continental arid. Because of its unique geographical location and special topography, the climate in the basin has been dry (average temperature of 10.7 °C). In the past 50 years, the average temperature of the basin has increased at a rate of 0.276 °C/10a with less rainfall (average annual precipitation is 17.4–42.8 mm), and in the past 50 years, the precipitation in the basin has increased at a rate of 6.05 mm/10a, and the evaporation intensity is large (the evaporation is 1800–2900 mm) [34], which indicates that the climate in the basin is affected by global changes and is developing from warm dry to warm wet. The Basin has an average altitude of between 800 and 2000 m and features a diversity of geological landform units, such as deserts, oases, mountains, and plateaus. There are also 114 rivers of various sizes crisscrossing the region, the major ones of which are the Tarim, the Aksu, the Kaidu-Kongqi, the Yarkant, and Hotan. The rivers in the basin are mainly distributed around the Taklimakan Desert in a ring shape, and the water supply mainly depends on the melting of ice and snow.

The main administrative regions within the Basin include Bayingoleng Mongolian Autonomous Prefecture, Aksu Prefecture, Hotan Prefecture, Kashgar Prefecture, and Kizilsu Kirgiz Autonomous Prefecture. The rivers in the basin are mainly distributed around the Taklimakan Desert in a ring shape, and the water supply mainly depends on the melting of ice and snow.

### 2.2. Methods

#### 2.2.1. Data Collection and Processing

The present research uses the four phases of land use status data from 1990, 2000, 2010, and 2018, sourced from the Environmental Science Data Center of the Chinese Academy of Sciences (http://www.resdc.cn/). The spatial resolution is 30 m, and the classification accuracy of the data is over 90% [35]. Based on the national land use classification system, the land use types in the study area are

divided into cultivated land, forest land, grassland, water area, construction land, and unused land. The socio-economic data and demographic data involved in the research are derived from 1990–2018 (Xinjiang Statistical Yearbook), (Production and Construction Corps Statistical Yearbook), and the website of Xinjiang Uygur Autonomous Region Statistics Bureau (http://www.xjtj.gov.cn/).

### 2.2.2. Single Dynamic Degree of Land Use

The indicator of single dynamic degree of land use can quantitatively describe the rate and magnitude of change in different land use types in the study area within a certain time range [36]. Based on land type data for the Tarim River Basin in 1990, 2000, 2010, and 2018, the 28-year land use/cover changes in the Basin were analyzed as follows:

$$K = \frac{U_j - U_i}{U_i} \times \frac{1}{t} \times 100\% \tag{1}$$

where $K$ is the dynamic degree of the land use type from $j$ to $i$, $U_i$ and $U_j$, respectively, representing the area of the land use type at the beginning and end of the study period, and $t$ is the research duration.

### 2.2.3. Ecosystem Service Value (ESV) Calculation

To infer the value equivalent of China's ecosystem services, this study uses the Costanza evaluation model proposed by Xie and others, based on the actual situation in China [37]. Farmland and difficult-to-use land is divided into cultivated land and unused land, respectively, while construction land has cultural and recreational ecological services only [38]. By revising according to the actual situation of the study area [39], the service value coefficient per unit area of the Tarim River Basin ecosystem has been determined (Table 1). According to the service value coefficient and the area of each land use type in the Basin, the ESV of the study area can be calculated using the following formulae:

$$ESV = \sum A_a \times VC_a ESV = \sum A_a \times VC_{ba} \tag{2}$$

where $ESV$ indicates ecosystem service value, $ESV_b$ denotes the ecosystem service value of item $b$, $A_a$ refers to the area of type $a$ land use in the study area, $VC_a$ represents the ecological service value coefficient of type $a$ land use, and $VC_{ba}$ is the value of the ecosystem service item $b$ of type $a$ land use.

**Table 1.** Ecosystem services value (ESV) coefficient of the Tarim River Basin (RMB/hm$^2$).

| Ecosystem Service Function | Land Use Type | | | | | |
|---|---|---|---|---|---|---|
| | Cultivated Land | Forest Land | Grass Land | Water | Construction Land | Unused Land |
| Gas regulation | 940.91 | 6586.37 | 1505.46 | 0.00 | 0.00 | 0.00 |
| Climate regulation | 1674.82 | 5080.92 | 1693.64 | 865.64 | 0.00 | 0.00 |
| Water conservation | 1129.09 | 6021.83 | 1505.46 | 38,351.5 | 0.00 | 56.45 |
| Soil formation and protection | 2747.46 | 7339.10 | 3669.55 | 18.82 | 0.00 | 37.64 |
| Waste disposal | 3086.19 | 2465.18 | 2465.18 | 34,211.5 | 0.00 | 18.82 |
| Biodiversity conservation | 1336.09 | 6134.74 | 2051.18 | 4685.73 | 0.00 | 639.82 |
| Food production | 1881.82 | 188.18 | 564.55 | 188.18 | 0.00 | 18.82 |
| Raw material production | 188.18 | 4892.73 | 94.09 | 18.82 | 0.00 | 0.00 |
| Entertainment culture | 18.82 | 2408.73 | 75.27 | 8167.10 | 82.60 | 18.82 |
| Total | 13,003.38 | 41,117.78 | 13,624.38 | 86,507.29 | 82.60 | 790.36 |

### 2.2.4. Ecosystem Service Change Index

The present study uses the Ecosystem Service change index to determine changes in ecosystem services [40]. To express the relative gain or loss of various ecosystem services, the calculation formula is:

$$ESCI_x = \frac{ES_{CUR_x} - ES_{HIS_s}}{ES_{HIS_s}} \tag{3}$$

where $ESCI_x$ represents a single ecosystem service change index and $ES_{HISs}$ and $ES_{CURx}$ correspond to ecosystem services in the beginning and end states, respectively. Furthermore, when $ESCI > 0$, there is a gain; when $ESCI < 0$, there is a loss; and when $ESCI = 0$, there is no change.

### 2.2.5. Spatial Autocorrelation Analysis

Based on Global Moran's *I* and Univariate Local Moran's *I* in the Geoda model, the agglomeration and abnormality of the spatial distribution patterns of ecosystem services are explored [41]. $G_i^*$ is used to examine the distribution characteristics of special values in the spatial changes of ESV, namely high-value agglomeration (hot spots) and low-value agglomeration (cold spots) [42]. The formulae are:

$$I_i = \frac{x_i - \overline{X}}{S_i^2} \sum_{j=1,j\neq i}^{n} w_{i,j}\left(x_j - \overline{X}\right) \tag{4}$$

$$S_i^2 = \frac{\sum_{j=1,j\neq i}^{n}\left(x_j - \overline{X}\right)^2}{n-1} \tag{5}$$

where $x_i$ is the attribute of element $i$, $\overline{X}$ is Average value of corresponding attributes, $W_{i,j}$ is the spatial weight between elements $i$ and $j$, $n$ is the total number of elements.

$$G_i^* = \frac{\sum_{j=1}^{n} w_{ij}x_j - X\sum_{i=1}^{n} w_{ij}}{S\sqrt{\left[n\sum_{j=1}^{n} w_{ij}^2 - \left(\sum_{j=1}^{n} w_{ij}\right)^2\right]/(n-1)}} \tag{6}$$

$$X = \frac{1}{n}\sum_{i=1}^{n} x_i, S = \sqrt{\frac{1}{n}\sum_{i=1}^{n} x_i^2 - (X)^{-2}} \tag{7}$$

where $x_j$ is the attribute value of element $j$, $w_{ij}$ is the spatial weight between elements $i$ and $j$ (adjacent is 1, non-adjacent is 0), $n$ is the total number of sample points. $X$ is mean, $S$ is standard deviation.

If the ESV change in a certain area is higher than the surrounding area, it becomes a hot spot with significant statistical significance, called ESV value-added hot spot, which means that the value of ecosystem services in this area has increased greatly. If the ESV change in a certain range is much lower than the surrounding area, it becomes a statistically significant cold spot, which is called the ESV loss cold spot area, which means that the ecosystem services value in this area decreases greatly [43].

## 3. Results and Analysis

### 3.1. Spatio-Temporal Change Characteristics of Land Use Types

The land use structure (Table 2) and current status (Figure 2) of the Tarim River Basin from 1990 to 2018 show that unused land is the largest and most widely distributed land use type in the study area, occupying a dominant position in the Basin's ecosystem. As of 2018, unused land accounted for 65.52% of the total area, followed by grassland (26.12%), water (2.67%), cultivated land (4.13%), and forest land (1.27%). Construction land area was the smallest, accounting for only 0.29% of the total area.

**Table 2.** Land use type in the Tarim River Basin (%).

| Land Use Type | 1990 | 2000 | 2010 | 2018 | 1990–2018 |
|---|---|---|---|---|---|
| Cultivated land | 2.37 | 2.61 | 3.08 | 4.13 | 1.76 |
| Forestland | 1.31 | 1.37 | 1.31 | 1.27 | −0.04 |
| Grassland | 27.99 | 27.12 | 26.87 | 26.12 | −1.86 |
| Water | 3.82 | 3.94 | 3.91 | 2.67 | −1.15 |
| Construction land | 0.16 | 0.14 | 0.16 | 0.29 | 0.13 |
| Unused land | 64.36 | 64.82 | 64.67 | 65.52 | 1.16 |

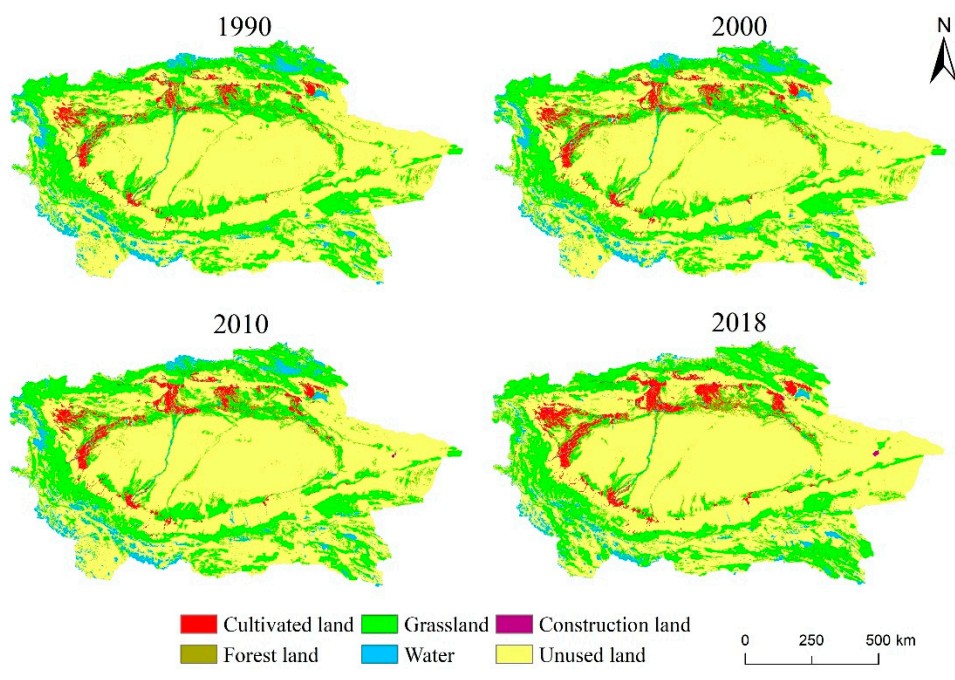

**Figure 2.** Status of land use in the Tarim River Basin.

During the study period, each land use type showed different characteristics of change. These were mainly manifested in the increase in cultivated land, construction land and unused land, and the decrease in forest land, grassland, and water area. Cultivated land showed the most obvious growth trend in the study area, increasing by $181.60 \times 10^4$ hm$^2$ over 28 years. The growth rate, which was the fastest from 2010 to 2018, had a dynamical degree of 4.30% (Table 3) and was mainly composed of grassland ($148.58 \times 10^4$ hm$^2$) and unused land ($49.53 \times 10^4$ hm$^2$) (Table 4). This change from grassland/unused land to cultivated land is mainly related to large-scale human reclamation activities in the study area.

**Table 3.** Dynamic degree of land use types in the Tarim River Basin (%).

|  | Cultivated Land | Woodland | Grassland | Water | Construction Land | Unused Land |
|---|---|---|---|---|---|---|
| 1990–2000 | 1.00 | 0.51 | −0.31 | 0.32 | −1.01 | 0.07 |
| 2000–2010 | 1.78 | −0.46 | −0.09 | −0.07 | 1.82 | −0.02 |
| 2010–2018 | 4.30 | −0.37 | −0.30 | −3.98 | 9.25 | 0.16 |
| 1990–2018 | 2.65 | −0.10 | −0.22 | −1.08 | 3.03 | 0.06 |

**Table 4.** Land use transfer matrix of the Tarim River Basin from 1990 to 2018 ($\times 10^4$ hm$^2$).

| Land Use Type | Cultivated Land | Woodland | Grassland | Water | Construction Land | Unused Land | Total | Transfer |
|---|---|---|---|---|---|---|---|---|
| Cultivated land | 195.01 | 10.62 | 17.02 | 2.54 | 13.14 | 6.23 | 244.56 | 49.55 |
| Woodland | 18.69 | 34.84 | 58.52 | 3.05 | 0.43 | 19.03 | 134.56 | 99.72 |
| Grassland | 148.58 | 68.16 | 1686.66 | 46.95 | 4.01 | 928.85 | 2883.21 | 1196.55 |
| Water | 4.32 | 2.27 | 78.47 | 151.92 | 0.65 | 152.57 | 390.20 | 238.28 |
| Construction land | 10.03 | 0.60 | 0.85 | 0.12 | 3.72 | 0.67 | 15.99 | 12.27 |
| Unused land | 49.53 | 13.92 | 854.99 | 67.84 | 7.61 | 5635.21 | 6629.10 | 993.89 |
| Total | 426.16 | 130.41 | 2696.51 | 272.42 | 29.56 | 6742.56 | 10,297.62 | |
| Transfer in | 231.15 | 95.57 | 1009.85 | 120.50 | 25.84 | 1107.35 | | |

During the period under study, construction land area in 1990 was $15.99 \times 10^4$ hm$^2$. This jumped to $29.56 \times 10^4$ hm$^2$ by 2018, representing an increase of $13.57 \times 10^4$ hm$^2$. As with cultivated land, the growth rate for construction land was the fastest in 2010–2018, with a dynamical degree of 9.25%.

The growth stemmed mainly from changes to cultivated land ($13.14 \times 10^4$ hm$^2$) and unused land ($7.61 \times 10^4$ hm$^2$).

The area of unused land in 1990 was $6629.10 \times 10^4$ hm$^2$. This increased slightly to $6742.56 \times 10^4$ hm$^2$ by 2018, representing a rise of 1.16%, with the fastest growth period occurring in 2010–2018. The dynamical degree was 0.16%, and the change in land used mainly came from grassland ($928.85 \times 10^4$ hm$^2$) and water area ($152.57 \times 10^4$ hm$^2$).

Desertification is also a factor in the change in land use area. Due to a decrease in precipitation and an increase in human activities and temperature, the desertification process accelerated. The area of forest land decreased $4.15 \times 10^4$ hm$^2$ over the 28 years of the study period, with the fastest decrease occurring between 2000 and 2010, at a dynamical degree of −0.37%. The main areas undergoing transformation were grassland ($58.52 \times 10^4$ hm$^2$), unused land ($18.69 \times 10^4$ hm$^2$), and cultivated land ($10.62 \times 10^4$ hm$^2$).

Grassland and water area also decreased by $168.70 \times 10^4$ hm$^2$ and $117.78 \times 10^4$ hm$^2$, respectively, with the fastest reductions occurring in 1990–2000 and 2010–2018, at dynamical degrees of −0.31% and −3.98%, respectively. Grassland mainly transformed into unused land ($928.85 \times 10^4$ hm$^2$) and cultivated land ($148.58 \times 10^4$ hm$^2$), while water mainly transformed into unused land ($152.57 \times 10^4$ hm$^2$) and grassland ($78.47 \times 10^4$ hm$^2$). These changes indicate a certain spatial synergy with the increase in cultivated land area and unused land. The large-scale reclamation and overgrazing led to a decrease in grassland area and an intensification of desertification. At the same time, the increase in agricultural irrigation water also contributed to the imbalance in the allocation of water resources, with water area showing an overall downward trend.

### 3.2. Temporal and Spatial Changes in ESV

### 3.2.1. Time Dimension Changes in ESV

Based on the first-level classification of land use, this study estimated the value of ecosystem services (Table 5) in the Tarim River Basin in 1990, 2000, 2010, and 2018. From 1990 to 2018, the overall ecosystem service value of the Tarim River Basin showed a downward trend, with a total decrease of 105.71 billion RMB (~12.11%). In 1990, 2000, and 2010, the value of ecosystem services in the study area was 872.88, 877.85, and 875.47 billion RMB, respectively, with a relatively flat trend. In 2018, however, the value of ecosystem services in the study area was 767.16 billion RMB, showing a clear downward trend.

**Table 5.** Changes in ESV of various land use types in the Tarim River Basin from 1990 to 2018/10$^8$ RMB.

| Land Use Type | | Cultivated Land | Woodland | Grassland | Water | Construction Land | Unused Land | Total |
|---|---|---|---|---|---|---|---|---|
| 1990 | ESV | 318.01 | 553.45 | 3930.10 | 3402.85 | 0.13 | 524.30 | 8728.84 |
| | Proportion % | 3.64% | 6.34% | 45.02% | 38.98% | 0.00% | 6.01% | 100% |
| 2000 | ESV | 349.89 | 581.82 | 3807.99 | 3510.64 | 0.12 | 528.04 | 8778.50 |
| | proportion % | 3.99% | 6.63% | 43.38% | 39.99% | 0.00% | 6.02% | 100% |
| 2010 | ESV | 412.23 | 555.21 | 3773.46 | 3486.94 | 0.14 | 526.78 | 8754.76 |
| | proportion % | 4.71% | 6.34% | 43.10% | 39.83% | 0.00% | 6.02% | 100% |
| 2018 | ESV | 554.15 | 538.56 | 3668.45 | 2376.53 | 0.24 | 533.71 | 7671.65 |
| | Proportion % | 7.21% | 7.01% | 47.72% | 30.91% | 0.00% | 6.96% | 100% |
| ESV changes from 1990 to 2018 | | 236.14 | −14.88 | −261.65 | −1026.32 | 0.11 | 9.42 | −1057.19 |
| ESV change rate from 1990 to 2018 | | 74.26% | −2.69% | −6.26% | −30.16% | 84.87% | 1.80% | −12.11% |

Based on the changes in ESV of various land types, it can be found that the ESV of water and grassland declined over the 28-year period, with a decrease of 102.63 and 26.16 billion RMB, respectively. Meanwhile, the ESV of cultivated land and unused land showed an upward trend, increasing by 23,614 and 942 million RMB, respectively. This indicates that the misuse of water resources broke the balance of the mutual transformation of ecosystem functions in the study area. Furthermore, the intensification of human activities and the acceleration of socioeconomic development were the main reasons for the overall decline in the value of ecosystem services in the Tarim River Basin.

Gas regulation, climate regulation, water conservation, and waste treatment constitute the regulation service functions of the study area. These four service functions showed a downward trend as a whole between 1990 and 2018. Among them, gas regulation, water conservation, and waste treatment service experienced a slight rebound from 1990 to 2000, but water conservation and waste treatment later showed a sharp decline from 2010 to 2018, with a decrease of 49.35 billion and 42.52 billion RMB, respectively (Table 6). Regulation services were mainly affected by cultivated land, woodland, grassland, and water. Since the grassland in the study area accounts for the largest proportion of the total area compared to the other three types of land, changes in the regulation services in the study area maintained a high correlation with changes in grassland area. So, for instance, the service functions of water conservation and waste treatment showed a rising trend followed by a falling trend, which is consistent with the aforementioned changes in grassland and water area.

**Table 6.** Changes in the value of ecosystem services in the Tarim River Basin from 1990 to 2018 ($10^8$ RMB).

| Type 1 | Type 2 | 1990 | | 2000 | | 2010 | | 2018 | |
|---|---|---|---|---|---|---|---|---|---|
| | | ESV | % | ESV | % | ESV | % | ESV | % |
| Regulation service | Gas regulation | 545.93 | 6.25% | 539.29 | 6.14% | 535.72 | 6.12% | 531.72 | 6.93% |
| | Climate regulation | 631.95 | 7.24% | 625.46 | 7.12% | 625.67 | 7.15% | 617.73 | 8.05% |
| | Water conservation | 2088.98 | 23.93% | 2130.46 | 24.27% | 2117.56 | 24.19% | 1624.06 | 21.17% |
| | Waste disposal | 2177.99 | 24.95% | 2207.88 | 25.15% | 2205.43 | 25.19% | 1780.14 | 23.20% |
| | Subtotal | 5444.85 | 62.37% | 5503.09 | 62.68% | 5484.38 | 62.65% | 4553.65 | 59.35% |
| Support service | Soil formation and protection | 1250.21 | 14.32% | 1229.32 | 14.00% | 1228.38 | 14.03% | 1227.19 | 16.00% |
| | Biodiversity conservation | 1315.69 | 15.07% | 1313.68 | 14.96% | 1308.61 | 14.95% | 1250.37 | 16.30% |
| | Subtotal | 2565.9 | 29.39% | 2543 | 28.96% | 2536.99 | 28.98% | 2477.56 | 32.30% |
| Provision of services | Food production | 231.29 | 2.65% | 231.3 | 2.63% | 238.69 | 2.73% | 252.55 | 3.29% |
| | Raw material production | 98.34 | 1.13% | 101.36 | 1.15% | 98.85 | 1.13% | 97.96 | 1.28% |
| | Subtotal | 329.63 | 3.78% | 332.66 | 3.78% | 337.54 | 3.86% | 350.51 | 4.57% |
| Cultural service | Entertainment culture | 388.47 | 4.45% | 399.76 | 4.55% | 395.85 | 4.52% | 289.94 | 3.78% |
| | total | 8728.84 | | 8778.5 | | 8754.76 | | 7671.65 | |

The two service functions of soil formation and protection and biodiversity conservation are collectively called support services. In the study area, these two services maintained a gentle downward trend from 1900 to 2018, mainly affected by cultivated land, woodland, and grassland. The combined effects of the decline in forest area volatility, the continuous expansion of cultivated land, and the ongoing decline in grassland were primarily responsible for the trend.

The service functions of food production and raw material production are supply services, and the service functions of entertainment and cultural services are cultural services. These two service functions—supply and cultural—account for only a small proportion of the ecosystem service value in the study area.

The supply service function is mainly affected by cultivated land, woodland, and grassland, with the expansion of cultivated land area and the decline of grassland increasing the output of crops and livestock products. Although the food production function charted a slight upward trend during 1990–2010, there was drastic growth from 2010 to 2018. The raw material production service function showed a modest upward trend from 1990 to 2000, but then maintained a gradual downward trend from 2000 to 2018. This change trend is similar to that of cultivated land and grassland. The changes in the supply of services are mainly due to the combined effects of the three types of land (cultivated land, woodland, and grassland), all of which showed upward trends.

The cultural service function showed an upward trend in 1990–2000, followed by a downward trend in 2000–2018 that was relatively sharp from 2010 to 2018. This service function was mainly

affected by woodland, grassland, and water, and their respective changes. The trend is consistent with the change trends pertaining to forest land and water area.

From the above, it is clear that the value of ecosystem services in the Tarim River Basin has generally followed a downward trend from 1990 to 2018, which is consistent with the change trend of regulation services. Although showing a slight upward trend prior to 2000, the ESV's downward trajectory emerged after 2000 and experienced a sharper descent from 2010 to 2018.

### 3.2.2. Changing Characteristics in ESV's Spatial Dimension

This study uses ArcGIS 10.2 software to divide the study area into 10,641 grids measuring 10 km × 10 km each [44]. As shown in Figure 3, the area of each category in the grid is calculated, using the natural breakpoint method to serve the ecosystem of the study area. The profit and loss of value and ESV is then spatially displayed (Figure 3).

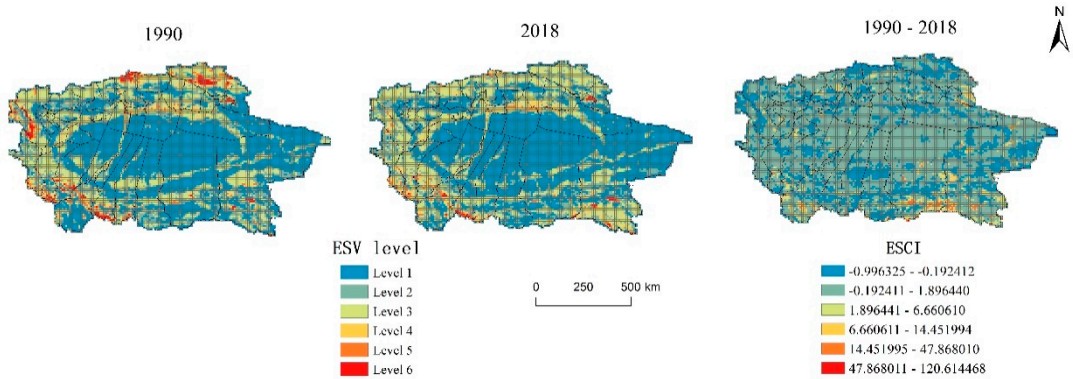

**Figure 3.** Spatial distribution characteristics of ESV in the Tarim River Basin.

Except for the northeastern part of the Tarim River Basin, the spatial distribution patterns of ESV for 1990–2018 shows that it is low in the central region and high in the surrounding area. This is largely consistent with the unique distribution of landform types in the study area. In the central Basin, a vast swathe of desert is unused land with low ESV. The surrounding plains area, however, which is hemmed by deserts and dotted with oases, is dominated by cultivated land, mountains, and large areas of grassland, and therefore has high ESV. In general, the high-value areas are mainly distributed in the grassland and water located in the north, west, and southwest of the study area.

Comparing the changes in the spatial distribution of ESV in 1990 and 2018, the high-value areas showed a downward trend, whereas the low-value areas expanded year by year. On the whole, ESV in the study area exhibited a significant downward trend.

According to the changes in profit and loss (ESCI) from 1990 to 2018, the ESV of the Tarim River Basin has reduced significantly, mainly due to the acceleration of human activities and urban economic development. Land use types with higher ESV (such as grassland and water) have been converted into land types with lower ESV (such as unused land). In the southeastern part of the study area, there is a plate-like distribution of gain areas, which is mainly due to the increase in water and grassland area; there was also some point-like gain areas in the northeastern part of the study area, which were mainly due to partial ecological restoration of grassland and water. Meanwhile, the ESV gain area in the western part of the study area was mainly due to the increase in grassland coverage in some regions due to the development of water resources.

### 3.3. Analysis of Spatial Correlation of ESV

### 3.3.1. Spatial Autocorrelation Analysis

This paper used the global autocorrelation analysis under the Geoda model to calculate the spatial patterns characteristics of the ecosystem service value of the Tarim River Basin in four periods from

1990 to 2018 (Figure 4). According to the calculation results, the ESV distribution in the study area has a very significant spatial correlation, where the regions with high ESV values were concentrated in spatial distribution, and vice-versa. The global Moran's I index in the four periods of 1900, 2000, 2010, and 2018 showed a downward trend year by year, with a maximum value of 0.6613 in 1990, indicating that the spatial correlation of ESV declined year by year for grass and water. At the same time, the higher value areas also decreased year by year.

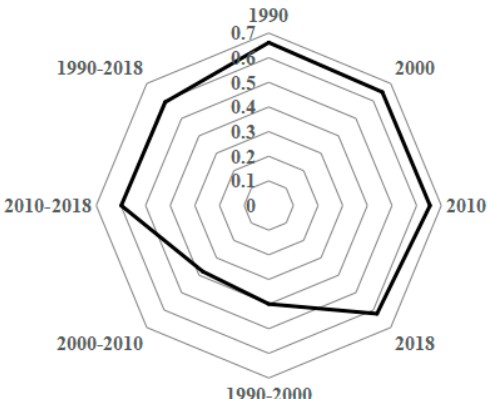

**Figure 4.** Changes in the Moran index in the Tarim River Basin from 1990 to 2018.

By analyzing the correlation of the dynamic changes in ESV in the study area, a clear trend emerges that shows a decline followed by an increase, with a maximum value of 0.5985 reached in 2010–2018. This phenomenon is mainly due to the drastic changes in land use type in 2010–2018. However, a similar spatial agglomeration phenomenon in the study area is not clear. A local Moran's I index is used to analyze and obtain the LISA (Local Indicators of Spatial Association) cluster map (Figure 5).

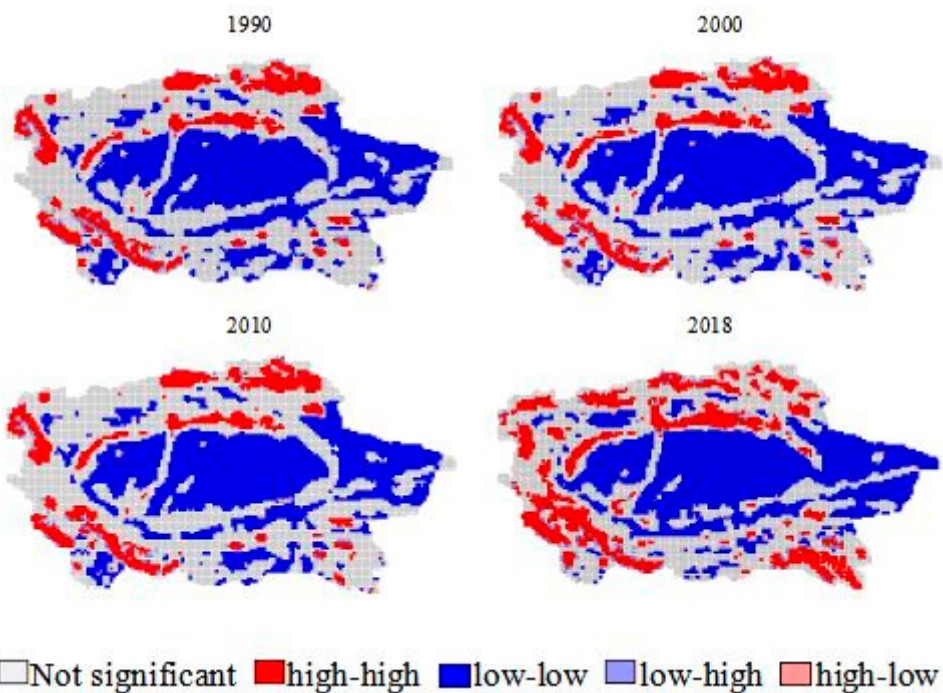

**Figure 5.** LISA cluster map of Tarim River Basin ESV from 1990 to 2018.

The main spatial clustering types of ESV in the study area are high-high, low-high, and low-low. The high-high ESV areas from 1990 to 2018 were mainly distributed in the northern, western, and southern

areas of the Basin, along with a small portion in the southeast. These areas mainly comprised grassland, cultivated land, and water were rich in resources and intensive human activities and had high levels of socio-economic development. In contrast, low-high areas were mainly distributed in the western and southeastern areas and were mostly unused land and grassland. In these regions, there were few land resources with high ESV. Finally, low-low ESV areas were mainly distributed in the middle and northeastern portions of Basin, where the main land types were unused land with low ESV. In these regions, the ecological environment is fragile, making it difficult to develop and utilize the available resources.

### 3.3.2. Analysis of Cold and Hot Spots

With the aid of the analysis tool ArcGIS 10.2 software, statistically significant hot and cold spots were selected, with a confidence of more than 90% in the obtained results. The dynamic changes of ESV in the study area are spatially expressed in Figure 6. The distribution of value-added hot spots in the Tarim River Basin from 1990 to 2000 and 2000 to 2010 was relatively discrete and mainly occurred in the oasis–desert interlace and parts of the south. This was primarily due to the conversion of unused land to cultivated land and the increase in water area. ESV loss cold spots were mostly distributed in the southwest portion of the study area, due mainly to the conversion of grassland to cultivated land and unused land. The ESV gain hot spots in 2010–2018 were distributed in the southeast portion of the Basin. This was due to the rise in temperature and subsequent accelerated snow melting in this area, which led to an increase in water resources. As well, the grassland ecological environment caused the growth of grassland area and water area to increase ESV in this region. Meanwhile, ESV loss cold spots were mainly distributed in the northern and western regions of the study area, caused by the ramping up of human activities and large-scale reclamation. Activities such as grazing also caused the conversion of land types with high ESV (e.g., grassland) to land types with low ESV (e.g., unused land).

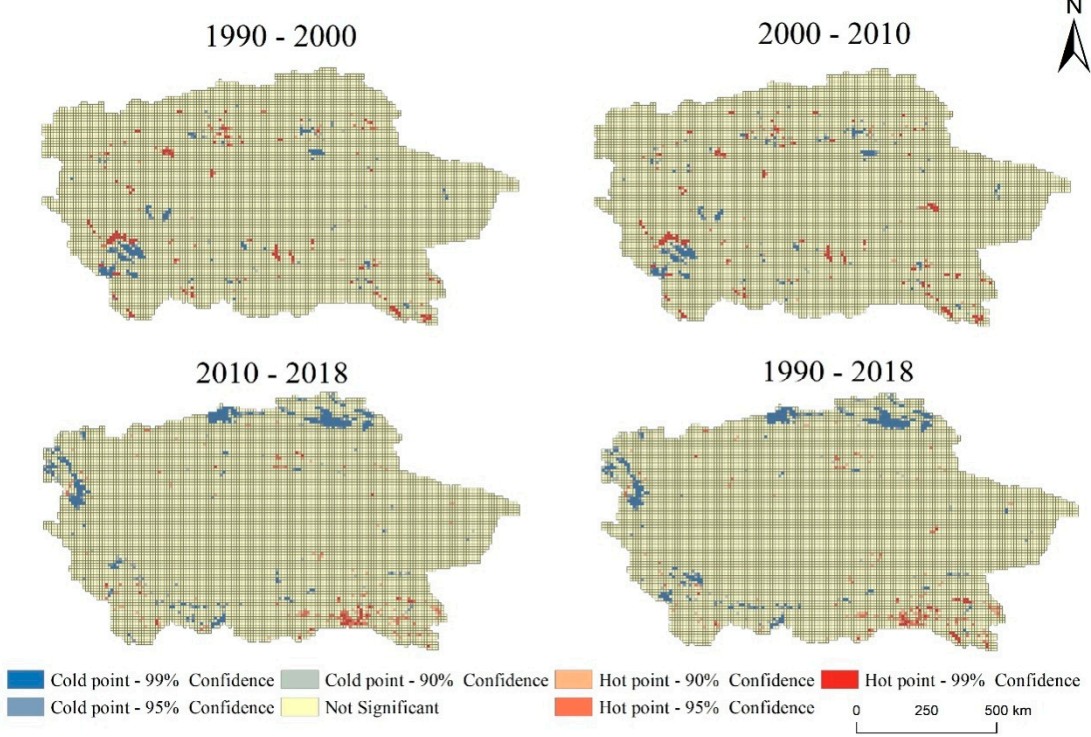

**Figure 6.** Analysis of ESV hot and cold spots in the Tarim River Basin.

On the whole, the changes in hot and cold spots from 1990 to 2018 are similar to the changes in ESV gains and losses in the study area in terms of spatial distribution characteristics. The value-added

hot spots of ESV were concentrated in the southeast of the study area and were mainly due to the presence of grassland and water. The cold spots of ESV loss were primarily distributed in the northern and western regions of the study area and were caused by the conversion of grassland and water areas to cultivated land and unused land. Based on the distribution area of hot and cold ESV changes in the study area, it can be seen that the ESV of the Tarim River Basin from 1990 to 2018 was in a state of loss.

## 4. Discussion

The Tarim River Basin is an ecologically fragile area in northwestern China [45]. Since 1990, various land use types in the Basin have undergone dramatic transformations. The study area is dominated by grassland and unused land, but due to the combined effect of global warming and the needs of social and economic development, a large amount of grassland has been transformed into cultivated land and unused land. The area change of the water in the study area showed an increase followed by a decrease, with an overall declining trend. The decline was the fastest from 2010 to 2018. The water areas were mostly turned into unused land due to the misuse and misallocation of water resources. The conversion of ESV high-value land types to low-value land types is one of the main reasons for the decline in the value of ecosystem services in the study area since the 1990s. This shows that while developing the economy in the Tarim River Basin, attention should be paid to the impact of a large reduction in the area of grassland on the fragile ecological environment, and that the increase in cultivated area in the upper reaches of the Tarim River and the increase in water conservancy facilities have caused many ecological problems, including the waste of water resources and the shortage of ecological water supply in the downstream [46], etc. While vigorously promoting people's living standards, we must also pay attention to improving the quality of the living environment. The increase and protection of ecological land will alleviate the contradiction between human and land to a certain extent and improve the overall ecosystem service value of the Tarim River Basin. Li [47] believes that the protection and increase in ecological land may require high upfront investment in the short term, but from a long-term perspective, both humans and the environment can obtain higher benefits. Decision makers should emphatically consider the trade-offs between land market returns, ecological environment restoration, and biodiversity conservation in the allocation of ecosystem services [48].

By comparing the change characteristics of ESV between the study area and other areas in XinJiang, it was clear that the change trend is not the same. For example, the value of ecosystem services in the Yanqi Basin showed a fluctuating downward trend from 1990 to 2010 [49], while the ESV of the northern edge of the Tarim River Basin showed a continuous decline from 1994 to 2016 [50]. Taking XinJiang as the scale, the value of ecosystem services from 2007 to 2016 generally showed growth [51]. The dissimilarities in the value of ecosystem services across various regions is mainly caused by the differences in land use structure, topography, time scale, and study area scale.

The results of the present study indicate that the value of the ecosystem services in the Tarim River Basin showed an upward trend during 1990–2000 but a downward trend during 2000–2018. Moreover, these results differ from those of some other scholars for the same region. Bai et al. estimated that the value of ecosystem services in the mainstream of the Tarim River from 2005 to 2010 showed an upward trend based on the monetary value evaluation method [52]. Similarly, Huang et al. estimated the Tarim River Basin's ESV based on the equivalent factor of China's terrestrial ecosystem service value formulated by Xie and others, showing an upward trend from 1994 to 2005 [53]. Therefore, when the equivalent factor method evaluates the value of ecosystem services, whether the equivalent factor is corrected according to the actual situation of the study area, the selection of the equivalent factor correction method is different from the parameter selection is one of the main reasons that the same evaluation method is used, but the evaluation results are different. Moreover, the selection of the study period will also cause differences in the evaluation results of ecosystem services.

Currently, there is no effective method for exploring the impact of land use change on the value of ecosystem services based on a spatial scale. This paper attempted to use the spatial autocorrelation analysis method based on grid cells to better express the response of ESV to land use change from a

spatial perspective. It also used the ecosystem service value LISA cluster map and heat map to show the ESV distribution change features, thus providing a possible way to investigate the impact of land use evolution on ecosystem services in the future. However, due to limitations in scope, the present study did not consider the natural and socio-economic driving factors behind the changes in watershed land use or in ESV, and so the spatial correlation is not clear. This will be the focus of a future study.

The Tarim River Basin is located in the core area of the Silk Road Economic Belt. The increasingly degraded ecological and environmental problems have attracted the attention of the Chinese government and the international community. As an important component of the "mountain–oasis–desert" coupling system, the oasis has its unique characteristics, evolution process, and trends. In recent years, under the background of global changes and the intensification of human activities, the ecological environment of the Tarim River Basin has become more fragile, and the contradiction between man and land has become more intense, which has seriously hindered the progress of the ecological civilization construction of the basin. This paper takes the temporal and spatial changes of ecosystem service value as the starting point, conducts a ground-breaking assessment of the ecological assets of the basin on time and space scales, and uses intuitive changes in ecological assets to measure the current status of the ecological environment of the basin to provide policy-makers with more popular theoretical support when formulating ecological restoration policies, and the research on the dynamic change of ecosystem service value on the spatial scale provides basic support for the formulation of ecological restoration policies in the division of regions to ensure the implementation of policies in accordance with local conditions.

## 5. Conclusions

By using land use data for the Tarim River Basin from 1990 to 2018, this paper looked at the evolution of land use patterns and how they impacted the value of ecosystem services. The study area was divided into grid units, applying the modified value-equivalent factor method per unit area. The main conclusions are as follows:

Unused land, grassland, and cultivated land are the land use types that account for a relatively large area (85%) of the Tarim River Basin. During the study period (1990–2018), the area of cultivated land in the Basin showed a continuous growth trend, and the unused land, water area, and construction land exhibited a fluctuating growth trend. Meanwhile, grassland charted a continuous decline, and water displayed a fluctuating decreasing trend. Among the different land use types, construction land had the highest increase (74%), followed by cultivated land; unused land had the lowest increase. Water was the land use type with the largest decrease in area, followed by unused land and forest land.

The ESV of the study area showed a slight upward trend from 1990 to 2000 and a downward trend from 2000 to 2018. The overall watershed ESV displayed a downward trend. Among them, the ESV declined sharply from 2010 to 2018. During this period, the grassland area in the watershed intensified, and the ESV high-value land use types frequently turned to low-value land use types.

Spatially, the value-added areas of ESV in the study area from 1990 to 2018 were mainly concentrated in the southeast, while areas of ESV loss were mainly distributed in the northern and western portions. Overall, the study results indicated that the ESV of the Tarim River Basin is in a state of loss.

**Author Contributions:** Y.W. conceived the study design, S.Z., X.C., H.Z. and R.S. implemented the field research collected and analyzed the field data, Y.W. wrote the paper with the help of S.Z., X.C., H.Z. and Z.L. All authors have read and agreed to the published version of the manuscript.

**Funding:** This research was funded by the National Key Research and Development Program (Grant No.: 2019YFA0606902) and National Natural Science Foundation of China (Grant No.: 41661015).

**Conflicts of Interest:** The authors declare they have no conflict of interest.

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
