# Peer review of "Spatiotemporal Evolution Characteristics in Ecosystem Service Values Based on Land Use/Cover Change in the Tarim River Basin, China"

_sustainability, doi:10.3390/su12187759_

Round 1
Reviewer 1 Report
Line 16 and 17. There is a contradiction about the water areas.
English needs to be corrected, plural not use when it should be patterns instead of pattern (line 58); making the basin ecological environment more fragile (Line 66); Paragraph line 69 to 74 needs to be edited for better comprehension
Author Response
Dear Reviewer:
On behalf of my co-authors, we thank you very much for giving us an opportunity to revise our manuscript, we appreciate editor and reviewers very much for their positive and constructive comments and suggestions on our manuscript entitled “Spatiotemporal Evolution Characteristics in Ecosystem Service Values Based on Land Use/Cover Change in the Tarim River Basin, China”(ID: sustainability-911857).
We have studied reviewer’s comments carefully and have made revision which marked in red in the paper. We have tried our best to revise our manuscript according to the comments. Attached please find the revised version, which we would like to submit for your kind consideration.
Thank you and best regards,
Yours sincerely

Reviewer 2 Report
The paper looks fine enough to be ready for being published.
Author Response
Dear Reviewer:
Thanks very much for your kind work and consideration on publication of our paper. On behalf of my co-authors, we would like to express our great appreciation to editor and reviewers.
Thank you and best regards.
Yours sincerely
Reviewer 3 Report
Introduction
Lines 69-74. Explain also what is the main hypothesis to demonstrate and why you have decided to use the images selected.
2.1. Description of the study area
Lines 83-85: More concrete climatic and meteorological information would be needed, to support subsequent interpretations about climate change and regional differences. At least information about anomalies from average in temperatures and precipitation in the study period would be very interesting.
Line 90: Include information about the landscapes of the study area and their spatial organization, as well as about the location and origin of the water resources.
2.2 Methods
Line 96. Include the complete date (month and day) of the images Landsat used, to assess a potential influence of seasonality in the changes.
- Results and analysis
Lines 164-166 and table 2, please check the percentages for 2018 in the text and in table 2.
- Discussion
Lines 365-376: It is difficult to understand the meaning of this paragraph. Probably because a drafting problem
Lines 392-394: This is an auto-evaluation of the study and should be explained deeply and more broadly
Author Response

(The authors gave the same response as above.)

Reviewer 4 Report
Authors have taken significant effort to investigate the watershed land use and ecosystem services value (ESV) space-time evolution characteristics in the Tarim River Basin in China’s arid northwest and authors have obtained interesting results and results were described sufficiently. However, address the following points:
Compare the current evaluation model and methods with the previous methods. And described advantages of the current model.
Explain the usefulness of this study.
Author Response

(The authors gave the same response as above.)
